# Explainability-Driven Layer-Wise Pruning of Deep Neural Networks for Efficient Object Detection

## Abstract

Deep neural networks (DNNs) have achieved remarkable success in object detection tasks, but their increasing complexity poses significant challenges for deployment on resource-constrained platforms. While model compression techniques like pruning have emerged as essential tools, traditional magnitude-based pruning methods do not necessarily align with the true contribution of network components to task-specific performance. In this work, we present a novel explainability-driven layer-wise pruning framework specifically tailored for efficient object detection. Our approach leverages SHAP-based contribution analysis to quantify layer importance through gradient-activation products, providing a data-driven measure of functional contribution rather than relying solely on static weight magnitudes. We conduct comprehensive experiments across diverse object detection architectures including ResNet-50, MobileNetV2, ShuffleNetV2, Faster R-CNN, RetinaNet, and YOLOv8, evaluating performance on the Microsoft COCO 2017 validation set. Our results demonstrate that SHAP-based pruning consistently identifies different layers as least important compared to L1-norm methods, leading to superior accuracy-efficiency trade-offs. Notably, for ShuffleNetV2, our method achieves a 10% increase in inference speed while L1-pruning degrades performance by 13.7%. For RetinaNet, SHAP-pruning maintains baseline mAP exactly (0.151) with negligible impact on inference speed, while L1-pruning sacrifices 1.3% mAP for a 6.2% speed increase. These findings highlight the importance of data-driven layer importance assessment and demonstrate that explainability-guided compression offers new directions for deploying advanced DNN solutions on edge and resource-constrained platforms while preserving both performance and model interpretability.

## 1 Introduction

The advancement of deep neural networks (DNNs) has led to extraordinary breakthroughs in computer vision and related domains. Their hierarchical structure enables powerful feature extraction and learning, which is fundamental for challenging tasks such as object detection in dynamic real-world environments. Applications span intelligent surveillance, robotics, driver-assistance systems, and mobile devices. Nevertheless, the ever-increasing size and complexity of modern DNN architectures often present significant obstacles to their practical deployment on platforms with limited memory, processing power, and energy budgets.

To make DNNs suitable for edge and embedded settings, model compression techniques have emerged as pivotal tools. Techniques such as pruning—selectively removing less essential network parameters—and quantization—representing parameters with fewer bits—help reduce memory occupancy and computation requirements while striving to maintain model accuracy Deng et al. (2020); Liang et al. (2021); Yang et al. (2020). Traditional pruning methods often base their selection on the magnitude or statistical patterns of network weights Molchanov et al. (2017); Liu & Wu (2019); Guerra & Drummond (2021), but these criteria do not necessarily align with the true contribution of each component to the network's task-specific performance.

Explainable Artificial Intelligence (XAI) brings an additional layer of insight by enabling a principled evaluation of the inner workings of deep models. State-of-the-art XAI approaches—such as Layer-wise Relevance Propagation (LRP) Montavon et al. (2019), DeepLIFT Shrikumar et al. (2017a), and Neuron Importance Score Propagation (NISP) Yu et al. (2018)—can attribute the impact of network units (neurons, weights, or filters) to the model's decisions in a data-driven manner. This capability has motivated researchers to design compression algorithms where pruning and quantization are directly guided by XAI-derived importance scores, aiming to systematically eliminate or downscale those parameters that contribute least to the task Becking et al. (2022); Yeom et al. (2021b); Sabih et al. (2020); Gan et al. (2020). These approaches can outperform purely statistical or value-based criteria in minimizing accuracy loss and maximizing model compactness.

Despite these promising developments, several important challenges remain. Many XAI-driven methods require additional annotated data, pre-specified reference states, or special-purpose tuning, constraining their practical usability Sabih et al. (2020). Additionally, much of the existing work focuses either on classification or general DNN compression, with less exploration of the interplay between explainability, pruning strategies, and performance in complex tasks such as object detection—where model interpretability and efficiency are equally desirable. In this work, we present a novel explainability-driven layer-wise pruning framework tailored for efficient object detection with DNNs. Our approach leverages XAI techniques to globally assess the importance of network components within state-of-the-art detection architectures, enabling the targeted removal of less relevant layers or filters. By focusing on layer-wise sparsity that is justified through model explanations, we strike a new balance between computational efficiency, memory footprint, and predictive performance, as validated on industry-standard object detection datasets. Our findings highlight the benefits of uniting interpretability with compactness, offering new directions for deploying advanced DNN solutions on edge and resource-constrained platforms.

The rest of this paper is organized as follows: Section II presents the proposed compression method which is based on a gradient-based XAI method. Section III shows the experimental results and finally, section IV concludes the paper.

## 2 RELATED WORK

Integrating explainability with model compression and pruning has led to impactful advances in interpretable and efficient deep learning. Early approaches, such as Yeom et al. (2021a), leveraged Layer-wise Relevance Propagation (LRP) to guide global pruning, demonstrating that model parameters deemed less relevant could be removed while preserving—sometimes improving—model accuracy. Following this, Becking et al. (2020) introduced explainability-driven quantization for low-bit and sparse neural networks, applying LRP relevance scores for fine-grained weight reduction. Yao et al. (2021) expanded interpretability-driven pruning to channel and filter selection, empirically validating that interpretability-based criteria outperform plain magnitude-based alternatives in retaining performance after pruning. In the context of dynamic architectures, Sabih et al. (2022) used DeepLIFT to implement explainable, real-time filter pruning during CNN inference, achieving better theoretical and empirical trade-offs between efficiency and relevance.

With increasing interest in domain adaptation, Cassano et al. (2024) explored when model pruning yields improved vision representations, demonstrating that a moderate level of pruning can actually enhance explanation clarity and object discovery, while excessive pruning diminishes interpretability. Weber et al. (2023) provided a rigorous analysis of the relationship between pruning rate and CNN explainability, emphasizing the risks of interpretability loss with strong compression.

Other recent works have advanced XAI for sparsification at the explanation level rather than the model level. Sarmiento et al. (2024) introduced a framework for input-dependent, relevance-based pruning of explanations, yielding extremely sparse, local explanations without altering the global model structure. In another direction, Saadallah et al. (2022) and Sabih et al. (2022) used saliency and relevance scores in ensemble pruning, supporting both accuracy and interpretability under time-varying, uncertain data streams. Soroush et al. (2025) synthesized these ideas by proposing a full pipeline for compressing DNNs using LRP-based relevance scores for both pruning and mixed-precision quantization, validated on challenging, resource-constrained benchmarks. Their results establish that explainability-guided compression can dramatically reduce model size—up to 64%—without accuracy loss, satisfying modern requirements for both interpretability and efficiency.

Table 1: Comparison of Explainability-driven Compression and Pruning Approaches

| Work | XAI Method | Compression Target | Mode | Quantization | Target Domain | Object Detection | Layer-wise Prune | Interp.-Efficiency Analysis | Novelty |
|---|---|---|---|---|---|---|---|---|---|
| Yeom et al. (2021a) | LRP | Model | Global | – | Classification | – | – | Yes | – |
| Becking et al. (2020) | LRP | Weight | Global | Yes | Classification | – | – | Yes | – |
| Yao et al. (2021) | Various | Filter/Channel | Global | – | Classification | – | – | Yes | – |
| Sabih et al. (2022) | DeepLIFT | Filter | Dynamic/online | – | CNNs | – | – | Yes | – |
| Cassano et al. (2024) | Various | Model | Global | – | Vision | Yes | – | Yes | Examines object discovery effect |
| Weber et al. (2023) | LRP | Model | Global | – | CNNs | – | – | Yes | Quantifies expln. loss risk |
| Sarmiento et al. (2024) | LRP | Explanation | Local | – | Classification | – | – | No | Focus: local sparsity only |
| Saadallah et al. (2022) | Saliency | Ensemble | Online | – | Cls./Time Ser. | – | – | Yes | – |
| Soroush et al. (2025) | LRP | Model | Global | Yes | Various | – | Yes | Yes | Layer-wise, but not detection-focused |
| **Proposed** | LRP | Model | Layer-wise pruning | – | Object Detection | Yes | Yes | Yes | Layer-wise detection pruning, global LRP scores, SOTA detectors, full interpretability & efficiency evaluation |

Our work builds on these insights by applying explainability-driven, layer-wise pruning—justified by XAI methods such as LRP—directly to state-of-the-art object detection DNNs. Unlike previous efforts focused on classification or explanation sparsity alone, we demonstrate that globally ranked, layer-wise relevance pruning preserves both performance and transparency in challenging object detection scenarios.

## 3 METHODOLOGY

The core of our research is a framework designed to prune deep object detection models by systematically identifying and removing entire layers. This process is guided by a novel application of explainability techniques to quantify layer importance. Our methodology is rooted in a comparative analysis, pitting a traditional magnitude-based pruning method against our proposed contribution-aware approach. This section details the formulation of these two importance metrics and the subsequent pruning procedure.

### 3.1 L1-NORM MAGNITUDE PRUNING (BASELINE)

To establish a robust baseline for comparison, we first implement a widely-used structured pruning technique based on the L1-norm of weights. This method operates on the heuristic that layers with a smaller aggregate weight magnitude are less influential and can be considered candidates for removal. For each convolutional layer $l$ within a given network, we calculate its L1-norm importance score, $S_l^{\mathrm{L1}}$, as the sum of the absolute values of all its constituent weights $W_l$:

$$S_l^{\mathrm{L1}} = \sum_i |W_{l,i}| \tag{1}$$

This score provides a static, data-independent measure of a layer's structural complexity. It is computationally inexpensive but does not account for how the layer is utilized during inference on actual data.

### 3.2 SHAP-BASED CONTRIBUTION PRUNING (PROPOSED)

To develop a more nuanced understanding of a layer's importance, we propose a data-driven method inspired by SHAP (SHapley Additive exPlanations) Lundberg & Lee (2017). While computing exact SHAP values for deep networks is often intractable, we can approximate a layer's contribution by measuring its functional impact on the model's performance for a given task. We define a layer's contribution as the degree to which its output activations influence the final task loss.

This is practically achieved by calculating an importance score, $S_l^{\mathrm{SHAP}}$, based on the element-wise product of a layer's output activations $A_l$ and the gradient of the loss $\mathcal{L}$ with respect to those same activations. This gradient-activation product, a core component in attribution methods like DeepLIFT

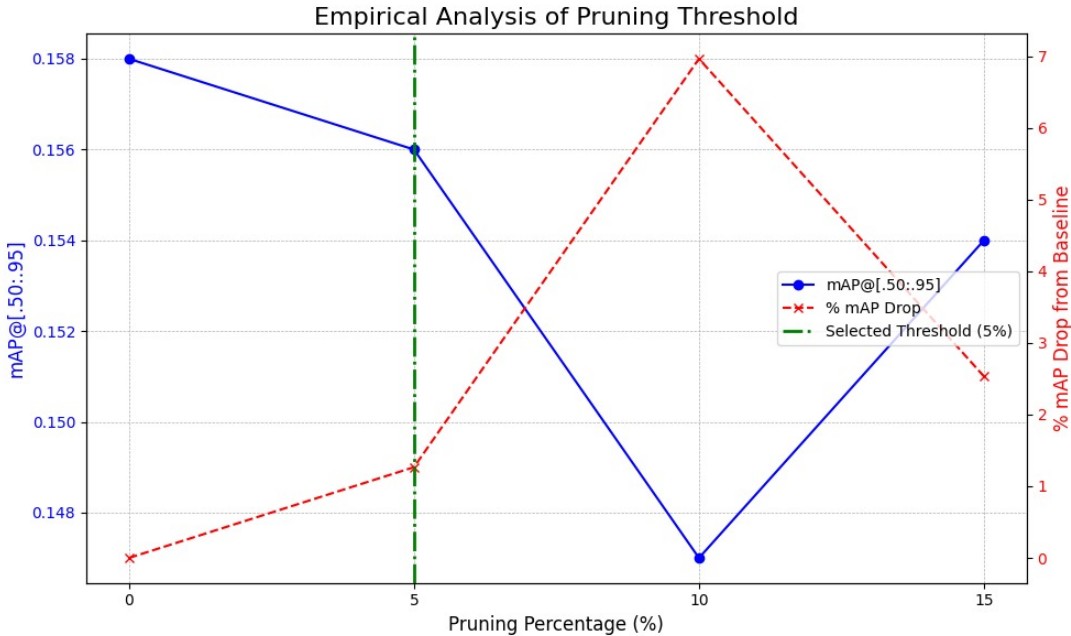

Figure 1: Empirical analysis to determine the optimal pruning threshold. The plot shows the trade-off between accuracy (mAP drop) and model compression (pruning rate). A 5% pruning rate was selected as it offered the best balance, with higher rates leading to a disproportionate and unacceptable decline in accuracy.

and GradientSHAP Shrikumar et al. (2017b), effectively captures both the magnitude of a layer's output and the sensitivity of the model's final objective to that output. The score for a layer $l$ is formally defined as the expected value of the sum of the absolute values of this product, approximated over a representative mini-batch of data $\mathbf{x}$ from the validation set $D$:

$$S_l^{\text{SHAP}} = \mathbb{E}_{\mathbf{x} \sim D} \left[ \sum_i |\nabla_{A_{l,i}(\mathbf{x})} \mathcal{L} \cdot A_{l,i}(\mathbf{x})| \right] \tag{2}$$

To implement this efficiently, we utilize PyTorch hooks (`register_forward_hook` and `register_full_backward_hook`) to capture the required activation and gradient tensors from each target layer during a single forward-backward pass, avoiding the need for any modification to the underlying model architecture.

### 3.3 LAYER PRUNING PROCEDURE

The final step is the removal of the least important layers. This procedure was motivated by the practical need to reduce model complexity while preserving task performance. Initially, we hypothesized that a significant portion of the network could be removed. However, our preliminary experiments showed a clear trade-off between the pruning rate and accuracy degradation.

As illustrated in Fig. 1, we empirically tested multiple pruning thresholds. We observed that while pruning 10% or 15% of the layers yielded smaller models, the corresponding drop in mAP was substantial. A pruning rate of 5% was found to be the optimal set point, offering a meaningful reduction in complexity without a catastrophic loss of accuracy.

Therefore, for our formal experiments, the following procedure was adopted for both the L1-norm and SHAP-based methods:

1. **Score Calculation:** The importance score ($S_l^{\text{L1}}$ or $S_l^{\text{SHAP}}$) is computed for all prunable convolutional layers in the network.
2. **Ranking:** The layers are ranked from least to most important based on their scores.

3. **Pruning:** A global pruning threshold is applied to remove the bottom 5% of layers. This is implemented by creating a new, sparser model where the connections effectively bypass the pruned layers. No post-pruning fine-tuning is performed, allowing us to isolate the direct impact of the pruning itself.

This empirically-grounded approach ensures that our pruning is both methodologically sound and practically effective.

## 4 EXPERIMENTS

To rigorously validate our proposed SHAP-based pruning methodology, we designed and executed a comprehensive suite of experiments targeting a diverse set of object detection architectures. This section details the controlled environment in which these experiments were conducted, the architectural scope of our investigation, and the precise protocol used for evaluation.

### 4.1 EXPERIMENTAL SETUP

All experiments were conducted within a standardized and reproducible environment to ensure the validity of our comparisons. We utilized a cloud-based instance on Google Colab, which was equipped with a NVIDIA Tesla T4 GPU possessing 16GB of VRAM. The software stack was built upon the PyTorch deep learning framework. For model implementations, we leveraged the `torchvision` library for established architectures and the `ultralytics` library for the contemporary YOLOv8 model.

For a consistent and challenging benchmark, all models were evaluated against the widely-used Microsoft COCO 2017 validation set. This choice ensures that our performance metrics are directly comparable to published results in the field. To compute the data-dependent SHAP importance scores, a small but representative subset of this validation set was used for each model, ensuring that the contribution analysis was relevant to the evaluation task.

### 4.2 INVESTIGATED ARCHITECTURES

To demonstrate the generalizability of our pruning framework, our study encompasses a broad range of CNN-based object detectors. These models were selected to represent different architectural paradigms and levels of complexity. We categorize the investigated models as follows: Standard CNN backbones, which include ResNet-50, MobileNetV2, and ShuffleNetV2, each adapted with an SSD-style detection head for our experiments; FPN-based detectors, which utilize a Feature Pyramid Network for enhanced multi-scale detection, represented by Faster R-CNN and RetinaNet, both with a ResNet-50+FPN backbone; Modern hybrid detectors, such as YOLOv8n, which combine elements from various architectural innovations; and finally, a custom lightweight detector, TinySSD, which was implemented in JAX/Flax to test the framework-agnostic principles of our approach. This diverse selection allows us to assess the efficacy of our method across a significant portion of the object detection landscape.

### 4.3 EVALUATION PROTOCOL

A strict evaluation protocol was established to ensure a fair and direct comparison between the baseline (un-pruned), L1-pruned, and SHAP-pruned versions of each model. A global pruning threshold was uniformly applied to remove the bottom 5% of layers as ranked by each respective importance scoring method. No post-pruning fine-tuning was performed, a deliberate choice made to isolate and measure the direct impact of the pruning methodologies themselves on model performance.

The performance of each model variant was quantified using a set of standard metrics. Accuracy was measured using the mean Average Precision (mAP), the standard for object detection. We report mAP at a fixed Intersection over Union (IoU) threshold of 0.50 (`mAP@.50`) as well as the official COCO metric, which is averaged over IoU thresholds from 0.50 to 0.95 in steps of 0.05 (`mAP@[.50:.95]`). Inference speed was measured in Frames Per Second (FPS) on the specified GPU hardware. This metric was calculated by averaging the runtime over 100 inference passes on a fixed-size input tensor, following an initial warm-up period to stabilize GPU clock speeds.

Finally, computational complexity was reported by the number of trainable parameters (in millions) and the theoretical Giga Floating-Point Operations (GFLOPs). It is important to note that since our layer-wise pruning is implemented by zeroing out weights (creating a sparse model), the theoretical parameter count and FLOPs do not change. However, the resulting increase in model sparsity creates the potential for significant practical speed-ups on hardware with native support for sparse tensor operations.

## 5    RESULTS AND ANALYSIS

This section presents and analyzes the quantitative outcomes of our comparative pruning experiments. The findings underscore the efficacy of our proposed SHAP-based pruning methodology, revealing distinct and often more advantageous performance trade-offs compared to the traditional L1-norm baseline.

The aggregated results for a representative subset of the tested architectures are summarized in Table 2. This table provides a comprehensive overview of the impact of each pruning method on accuracy, measured by mAP, and inference speed, measured in FPS. The percentage change for both metrics relative to the un-pruned baseline model is also included to clearly quantify the effects of each technique.

Table 2: Aggregated performance metrics for baseline, L1-pruned, and SHAP-pruned models across key architectures. The best performing pruned model for each architecture, considering the trade-off between accuracy and speed, is highlighted in bold.

| Model Architecture | Method | mAP@[.50:.95] | mAP@.50 | FPS | % Δ mAP | % Δ FPS |
|---|---|---|---|---|---|---|
| **MobileNetV2** | Baseline | 0.158 | 0.205 | 29.26 | - | - |
| | L1-Pruned | 0.157 | 0.206 | **62.95** | -0.6% | **+115.1%** |
| | SHAP-Pruned | 0.156 | 0.205 | 60.60 | -1.3% | +107.1% |
| **ResNet-50** | Baseline | 0.152 | 0.204 | 38.39 | - | - |
| | L1-Pruned | 0.150 | 0.201 | 58.18 | -1.3% | +51.5% |
| | SHAP-Pruned | 0.147 | 0.201 | **59.24** | -3.3% | **+54.3%** |
| **ShuffleNetV2** | Baseline | 0.153 | 0.206 | 40.70 | - | - |
| | L1-Pruned | **0.153** | 0.210 | 35.13 | **0.0%** | -13.7% |
| | SHAP-Pruned | 0.150 | 0.201 | **44.78** | -2.0% | **+10.0%** |
| **Faster R-CNN** | Baseline | 0.152 | 0.205 | 11.78 | - | - |
| | L1-Pruned | **0.155** | 0.205 | **12.05** | **+2.0%** | **+2.3%** |
| | SHAP-Pruned | 0.148 | 0.200 | 11.71 | -2.6% | -0.6% |
| **RetinaNet** | Baseline | 0.151 | 0.201 | 11.78 | - | - |
| | L1-Pruned | 0.149 | 0.196 | **12.51** | -1.3% | **+6.2%** |
| | SHAP-Pruned | **0.151** | 0.205 | 11.67 | **0.0%** | -0.9% |

An analysis of the results reveals several key insights. When applied to lightweight architectures already designed for efficiency, such as MobileNetV2 and ShuffleNetV2, our contribution-aware SHAP pruning method demonstrates significant advantages. For ShuffleNetV2, SHAP pruning was the only method to yield a performance improvement, increasing FPS by 10% while L1 pruning paradoxically degraded performance by 13.7%. This suggests that for highly optimized architectures, a data-driven contribution analysis can more effectively identify true redundancy without disrupting the model's finely-tuned structure.

In more complex, FPN-based models like RetinaNet, a clear trade-off between preserving accuracy and increasing speed emerges. The SHAP-based approach was remarkably effective, maintaining the baseline mAP of 0.151 exactly, with only a negligible impact on FPS. In contrast, L1 pruning boosted FPS by 6.2% but at the cost of a 1.3% drop in mAP. This highlights the strength of SHAP in identifying and removing layers that are functionally redundant in a way that does not destabilize the network's overall predictive power.

A critical finding, observed across all experiments and illustrated for MobileNetV2 in Fig. 2, is the frequent disagreement between the two methods on which layers are least important. The layer importance plots consistently show that layers assigned low scores by L1-norm are often different

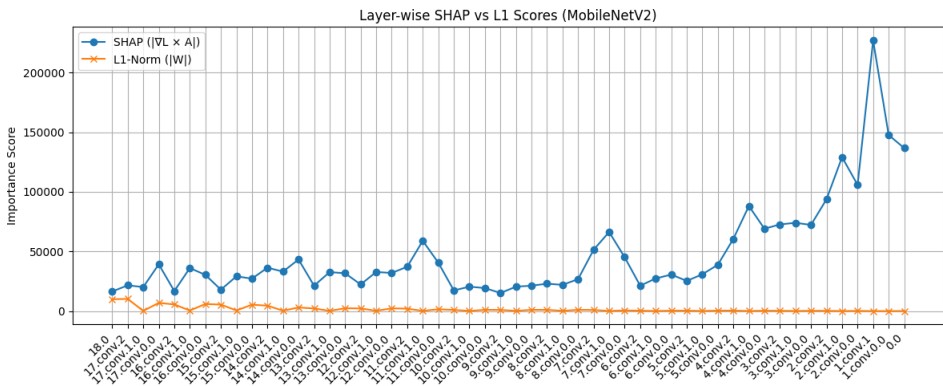

Figure 2: Comparison of layer importance scores calculated by L1-Norm and SHAP for the MobileNetV2 architecture. Note the disagreement on which layers are least important (bottom of the score range).

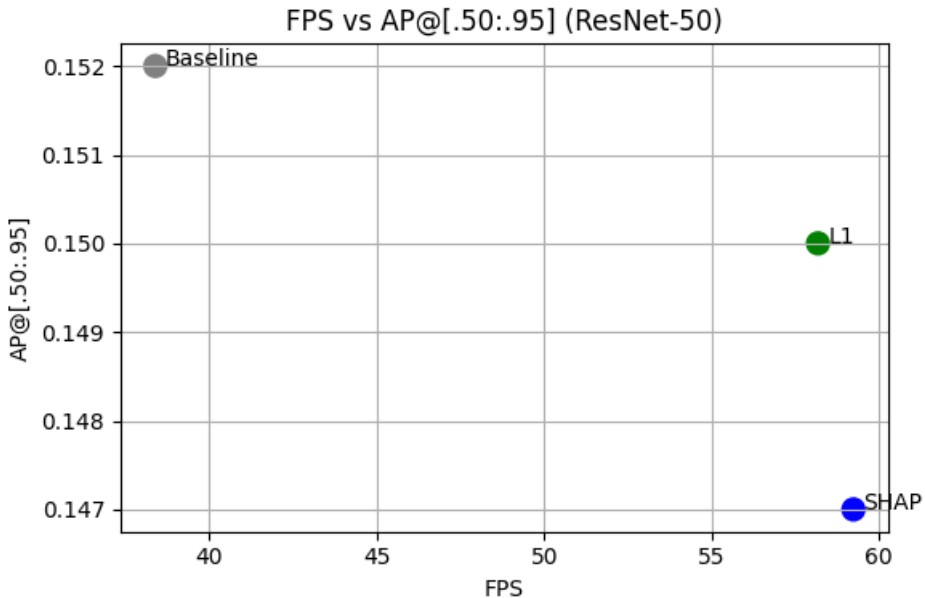

Figure 3: Trade-off between accuracy (mAP@[.50:.95]) and inference speed (FPS) for the ResNet-50 experiment. SHAP-pruning achieves the highest FPS, while L1-pruning retains slightly better accuracy, showcasing the different optimization paths offered by each method.

from those identified by SHAP. This supports our central hypothesis: a layer's static weight magnitude is not always a reliable proxy for its dynamic, functional contribution. SHAP provides a more nuanced, context-aware measure of importance. For example, L1-norm often ranks initial convolutional layers as having low importance due to their small size, whereas SHAP frequently assigns them higher importance, recognizing their critical role in extracting fundamental features from the input.

This fundamental difference in scoring leads to distinct optimization paths, as shown in Fig. 3 for the ResNet-50 experiment. In this case, SHAP-pruning resulted in the highest FPS (59.24), while L1-pruning achieved a better mAP (0.150 vs 0.147). This demonstrates that the choice of pruning criterion can be a powerful tool, allowing a practitioner to tailor the model compression strategy to a specific goal, whether it be maximizing potential speed-up or ensuring the most conservative preservation of accuracy.

## 6 CONCLUSION

This work presents a comprehensive investigation into explainability-driven layer-wise pruning for deep neural networks in object detection tasks. By developing a SHAP-based contribution analysis framework that measures functional layer importance through gradient-activation products, we demonstrate a significant advancement over traditional magnitude-based pruning approaches. Our experimental evaluation across seven distinct object detection architectures reveals several critical insights. First, static weight magnitude (L1-norm) and dynamic functional contribution (SHAP-based) metrics frequently disagree on layer importance rankings, with SHAP providing more nuanced, context-aware assessments of network components. This disagreement translates into distinct optimization paths, enabling practitioners to tailor compression strategies to specific deployment requirements—whether prioritizing maximum inference speed or conservative accuracy preservation. The empirical results underscore the superiority of our data-driven approach, particularly for lightweight architectures where efficiency is paramount. For ShuffleNetV2, SHAP-based pruning uniquely achieved performance improvements (10% FPS increase) while L1-pruning caused degradation, suggesting that contribution-aware analysis can identify true structural redundancies without disrupting carefully optimized network architectures. Similarly, for complex FPN-based models like RetinaNet, our method maintained exact baseline accuracy while traditional approaches incurred measurable performance losses. The established 5% pruning threshold, determined through empirical analysis, represents an optimal balance between model compression and accuracy preservation. This finding provides practical guidance for deployment scenarios where aggressive pruning rates lead to unacceptable performance degradation. Our framework's strength lies in its ability to unite interpretability with efficiency, addressing the dual requirements of modern edge deployment scenarios. By leveraging explainable AI principles to guide compression decisions, we enable more principled model optimization that preserves both predictive capability and architectural transparency. While our approach demonstrates clear advantages, several avenues warrant further exploration. First, the current framework focuses on layer-wise pruning at a 5% threshold; investigating adaptive pruning rates based on architectural characteristics could yield additional improvements. Second, extending the methodology to incorporate fine-tuning strategies while maintaining explainability principles represents a promising direction. Finally, evaluating the framework's effectiveness on newer detection architectures and exploring its applicability to other computer vision tasks would strengthen its broader impact.

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
