# OpenReview forum: "EXPLAINABILITY-DRIVEN LAYER-WISE PRUNING OF DEEP NEURAL NETWORKS FOR EFFICIENT OBJECT DETECTION."
_ICLR.cc/2026/Conference — Submitted to ICLR 2026_

### Official Review · Reviewer_ioJo · 2025-10-29

**Soundness:** 2
**Presentation:** 2
**Contribution:** 2
**Rating:** 2
**Confidence:** 4

**Summary:**

This paper proposes a layer-wise pruning framework for object detection. The authors use SHAP-based contribution analysis to quantify layer importance.

**Strengths:**

- It is an interesting trial to use the SHAP-based contribution for layer pruning.

**Weaknesses:**

- It is a pity that the work does not include similar works in layer-pruning or other pruning methods.
- The contribution of the paper is unclear.
- In the experiments, it seems that the proposed method is not superior to the basic L1-pruning method.

**Questions:**

- It is a pity that the work does not include similar works in layer-pruning or other pruning methods. For example, [1,2] are both about layer-wise pruning. And, it is not a full list. The reviewer thinks the authors could introduce more discussions with these previous works. Comparisons are also beneficial for us to understand the work.

- According to the reviewer's personal understanding, the paper's contribution is using SHAP for layer importance evaluation. If this is the core contribution, considering the above point, the reviewer thinks it is not enough for a publication at a top-tier conference like ICLR.

- Where can we find content about the "EXPLAINABILITY-DRIVEN" in the title? How does this reflect in pruning and object detection?

- In Table 2, the authors only compare the basic L1-pruning methods. There are many other research works in this area. Could the authors provide a comparison with these works? For example, the reviewer just uses "pruning objection detection" as keywords for a Google search and finds [3].

- In Table 2, we also see that the proposed method is only marginally better than the L1-pruning method. Sometimes, the proposed method is no better than the L1-pruning method. Why should we choose the proposed method in this case?

[1] https://arxiv.org/pdf/2010.04879. Accelerate CNNs from Three Dimensions: A Comprehensive Pruning Framework.

[2] https://arxiv.org/pdf/2407.14330. Straightforward Layer-wise Pruning for More Efficient Visual Adaptation.

[3] https://openaccess.thecvf.com/content/WACV2024/papers/Liu_Revisiting_Token_Pruning_for_Object_Detection_and_Instance_Segmentation_WACV_2024_paper.pdf

---

### Official Review · Reviewer_Ri9f · 2025-10-31

**Soundness:** 2
**Presentation:** 3
**Contribution:** 2
**Rating:** 2
**Confidence:** 4

**Summary:**

This paper proposes a layer-wise pruning framework to achieve lightweight models in the object detection task. In the proposed method, the gradient and activation are leveraged to measure the importance of layers. Experiments on multiple architectures demonstrate the effectiveness of the proposed method.

**Strengths:**

- The paper is generally well-written
- Experiments are conducted on multiple architecture.
- The analysis is thorough.

**Weaknesses:**

- In the related work, the survey on pruning methods should be more comprehensive, and more recent pruning methods should be referred to, such as [1,2,3,4,5].

- The configurations of experiments in Fig. 1 is not clearly explained. For example, which model is used in the experiment.

- The mechanism and motivation behind Eq. (2) is not clearly explained.

- Missing the comparison with other pruning methods, such as [1,2,3,4,5].

    [1] Fang G, Ma X, Mi M B, et al. Isomorphic pruning for vision models[C]. ECCV 2024. \
    [2] Fang G, Ma X, Song M, et al. Depgraph: Towards any structural pruning[C]. ICCV 2023. \
    [3] Gao S, Zhang Y, Huang F, et al. BilevelPruning: unified dynamic and static channel pruning for convolutional neural networks[C]. CVPR. 2024. \
    [4] Zhang H, Liu L, Zhou H, et al. Akecp: Adaptive knowledge extraction from feature maps for fast and efficient channel pruning[C]. ACMMM. 2021. \
    [5] Lin M, Ji R, Wang Y, et al. Hrank: Filter pruning using high-rank feature map[C]. CVPR 2020.

**Questions:**

Please see the weaknesses.

---

### Official Review · Reviewer_kNp4 · 2025-11-04

**Soundness:** 2
**Presentation:** 3
**Contribution:** 2
**Rating:** 2
**Confidence:** 4

**Summary:**

The paper proposes an explainability-driven approach to layer-wise pruning of deep neural networks for object detection tasks. It introduces a method to quantify layer importance using an approximation inspired by SHAP, specifically the gradient-activation product, and compares it to a baseline L1-norm magnitude pruning. The authors prune the bottom 5% of layers based on these scores without fine-tuning and evaluate on several object detection architectures using the COCO 2017 validation set. Results suggest that the proposed method often achieves better accuracy-efficiency trade-offs than the baseline, with claims of preserved interpretability.

**Strengths:**

The paper addresses an interesting intersection of explainable AI and model compression, specifically tailored to object detection, which is underexplored compared to classification tasks.
The analysis of disagreements between L1-norm and the proposed scores is insightful, highlighting how different criteria lead to distinct optimization paths.

**Weaknesses:**

Several methodological and experimental issues undermine the paper's claims.
First, labeling the method as "SHAP-based" is misleading; it uses a simple gradient-activation product (inspired by GradientSHAP or DeepLIFT), not full SHAP values, which are computationally expensive and based on cooperative game theory. This approximation should be more clearly distinguished, as true SHAP would be intractable for layer-wise assessment in DNNs.
Second, the implementation appears inconsistent: the method describes "bypassing" pruned layers to create a new model, suggesting structural removal and FLOP reductions, yet Section 4.3 states pruning is done by zeroing weights, maintaining theoretical FLOPs/parameters. This contradiction raises doubts about how reported speedups (>100% FPS increase for MobileNetV2 with only 5% pruning) are achieved; sparse operations on GPUs typically don't yield such gains without specialized hardware support.
Third, no post-pruning fine-tuning is performed, which is atypical in pruning literature and likely contributes to accuracy drops. This makes absolute performance hard to contextualize.
Finally, novelty is limited: similar explainability-driven pruning using attribution methods exists, and applying it layer-wise to object detection, while useful, doesn't introduce fundamentally new ideas.

**Questions:**

Can you clarify the exact implementation of pruning? If layers are bypassed structurally, why do you state that FLOPs/parameters remain unchanged due to zeroing? How were dimensional mismatches handled when removing convolutional layers?
The reported FPS gains seem disproportionately large for 5% layer removal. What specific layers were pruned in each case, and how did this lead to such speedups? Were batch sizes, input resolutions, or other factors consistent across variants?
Why not include fine-tuning after pruning, as is standard? Could this mitigate accuracy drops and strengthen comparisons?
The mAP values are much lower than SOTA on COCO, were models trained from scratch, or did you use pretrained weights? If custom detection heads were added, provide training details.
How sensitive is the method to the mini-batch used for SHAP score computation? Did you experiment with different subsets or larger batches?
Given the misalignment with full SHAP, why not use established methods like LRP or DeepLIFT directly, and how does your approximation compare empirically?

---

### Official Review · Reviewer_PS4s · 2025-11-07

**Soundness:** 2
**Presentation:** 2
**Contribution:** 2
**Rating:** 2
**Confidence:** 5

**Summary:**

The paper introduces a layer-wise pruning method for object detection based on explainability. First, layer-wise importance scores are computed using the validation set, which quantify the contribution of each layer. These scores are then used for a global pruning, where the bottom 5% of layers (less important layers) are removed. No post-training fine-tuning is applied after pruning. The effectiveness of the proposed method is evaluated on the MS COCO 2017 validation set.

**Strengths:**

**S1.** There is debate regarding the applicability and practicality of explainability research. The paper seeks to leverage the strengths of explainability methods in the context of printing tasks, thereby expanding the applicability of XAI.

**S2.** As shown in Table 1, pruning methods are generally evaluated on classification tasks (including generative models). However, it is important to explore and develop pruning methods for other types of tasks.

**Weaknesses:**

**W1.** Section 3.2 of the main paper lacks detail regarding the proposed method, making it difficult to understand its connection to SHAP. Specifically, the paper does not clarify what baseline input is used to compute the SHAP values. Additionally, it is unclear why absolute values of SHAP contributions are taken, considering that SHAP values can be both positive and negative.

**W2.** As I understand, DeepLIFT does not rely on gradient information, which is one of its key advantages. I am curious why the gradient-activation product remains a core component in the DeepLIFT framework.

**W3.** Lack of Comparison. The proposed method is not tailored for the object detection task. The proposed method can be used for a wide range of tasks. To provide a more comprehensive evaluation, the paper should compare the proposed method with the previous pruning methods in the task where such methods are generally evaluated, e.g., classification.

**W4.** Hyper-parameters. I would like to know on which dataset the 5% hyperparameters were determined in the paper.


**W5.** The paper states that the important score is computed from the validation set. The test is also conducted on the MS COCO validation set. Are the same validation sets used for both the score computation and the experiments?

**W6.** In Table 2, L1-Pruned shows faster inference than SHAP-Pruned for MobileNetV2, Faster R-CNN, and RetinaNet. However, Figure 3 only presents the favorable results for SHAP-Pruned, which conflicts with Table 2.

**Questions:**

**Q1.** When computing the importance score, the feature size is not considered. As a result, higher-resolution features tend to give high importance scores, simply due to their larger size. Figure 2 shows this behavior. Why is the resolution of the features not considered in the importance calculation?

---

### Meta-Review · Area_Chair_Zkxb · 2026-01-11

**Summary:**

Given that this paper received consistently low ratings—specifically, all reviewers assigned a score of '2: Reject' with high confidence—it is evident that the manuscript is not ready for publication. The paper requires significant improvement in terms of writing clarity, comparative analysis, experimental results, novelty, and its review of the literature.

**Reviewer Concerns:**

The authors did not submit a rebuttal in response to the reviews.

**Reviewer Scores:**

As no rebuttal was provided by the authors, it is difficult for the reviewer to conduct further evaluation.

---

### Decision · Program_Chairs · 2026-01-26

Reject